# Robust multipartite entanglement in dirty topological wires

Luca Pezzè[1] and Luca Lepori[1]

**1** QSTAR and INO-CNR and LENS, Largo Enrico Fermi 2, 50125 Firenze, Italy

December 7, 2022

## Abstract

Identifying and characterizing quantum phases in the presence of long range correlations and/or spatial disorder is, generally, a challenging and relevant task. Here, we study a generalization of the Kiteav chain with variable-range pairing and different site-dependence of the chemical potential, addressing commensurable and incommensurable modulations as well as Anderson disorder. In particular, we analyze multipartite entanglement (ME) in the ground state of dirty topological wires by studying the scaling of the quantum Fisher information (QFI) with the system's size. For nearest-neighbour pairing, the Heisenberg scaling of the QFI is found in one-to-one correspondence with topological phases hosting Majorana modes. For finite-range pairing, we overcome notable difficulties in characterising the system: in particular, we recognize long-range phases by the super-extensive scaling of the QFI and identify complex lobe-structured phase diagrams. The present work contributes to establish ME as a central quantity to study intriguing aspects of topological systems and testifies its robustness against spatial inhomogeneities.

# 1 Introduction

An essential property of topological phases of matter [1, 2] is their robustness against local perturbations [3, 4]. For instance, noninteracting symmetry-protected topological phases [5] are characterized by the presence of robust gapless boundary states. The number of such protected states is proportional to the topological invariant characterizing the system and plays the role of a global order parameter. The invariant chances only through a quantum phase transition closing and reopening an energy bulk gap and is thus immune to small but finite local perturbations. One of the most important examples of symmetry-protected topological systems is the Kitaev chain [6, 7]: a celebrated tight-binding model of one-dimensional spinless fermions exhibiting $p$-wave superconductivity. This prototype hosts non-local Majorana modes that are localized at the edges of an open chain and show non-Abelian exchange statistics under braiding [8, 9]. The intrinsic robustness of topological properties against local perturbations holds the promise to realize resilient quantum information processing [10–12], such as topologically-protected qubits and gate operations [13].

So far, the robustness of topological phases in the Kitaev chain has been mainly characterized by studying the fate of Majorana modes when including sufficiently strong local perturbations such as spatial inhomogeneities [14–21], eventually also in the presence of interaction [22–24] or long-range pairing [25–27]. However, besides the presence of edge states, the Kitaev model is also characterized by non-trivial quantum entanglement properties [28–34]. Is such entanglement robust against local perturbations? Is it as robust as the Majorana modes?

Generally speaking, the characterization of quantum phases and quantum phase transitions by entanglement-based approaches is an intriguing problem, at the frontier between quantum information [35, 36] and many-body physics [37–41]. The literature has mostly focused on bipartite entanglement, with witnesses such as the Von Neumann entropy [28, 29, 42–44], the entanglement spectrum [32, 45–49] and pairwise entanglement [50, 51], also in the presence of disorder [52–54]. Instead, multipartite entanglement (ME) has been much less studied [55, 56], while it captures a more complex entanglement structure than than identified by bipartite and pairwise entanglement. A prominent tool to analyze ME is the quantum Fisher information (QFI) [57–59], which is also also central in quantum metrology [60, 61]. The QFI has been investigated in topological [33, 62–66] as well as spin [67–72] and lattice [73, 74] systems, and used to characterize interesting many-body phenomena such as quantum criticality [75–77], quantum chaos [78, 79], quantum quenches [80, 81], scrambling [82] and thermalization [83, 84].

In this work, we investigate the stability of ME against spatial inhomogeneities in a generalization of the Kitaev chain where the superconducting pairing decays with distance as a power law [31, 85]. We consider commensurate and incommensurate quasiperiodic modulations of the chemical potential, as well as a uncorrelated on-site random disorder with Anderson distribution. These perturbations respect the Hamiltonian symmetries (charge conjugation and $Z_2$ number parity). We characterize various short-range (SR) and long-range (LR) phases by different scaling of the QFI with the system's size. We find that the super-extensive scaling of QFI is robust against inhomogeneities and holds up to modulation strengths strong enough to induce a quantum phase transition. In particular, inhomogeneities induce complex structure, such as re-entrances and lobes in interesting phase diagrams, also extending the topological phases in some cases. In the SR case, we observe a perfect agreement between the super-extensive scaling of the QFI and non-trivial topology phases identified by topological invariants [14–16]. Furthermore, tuning the pairing range can induce a transition from SR to LR phases that occurs without closing the mass gap: the SR-to-LR transition is captured by the change of scaling of the QFI, also in the presence of spatial inhomogeneities. In the LR case, where topological invariants are generally difficult to be defined, especially in the absence

of translational invariance, our methods are particularly relevant since they provide a clear characterzation of the phase diagram and the unprecedented identification of LR phases. On the general conceptual side, our study testifies, for the first time to the best of our knowledge, the robustness of spatial multipartite entanglement against spatial inhomogeneities. The impressive one-to-one correspondence between non-trivial topology and the scaling of the QFI – as shown in this manuscript – calls for applying the QFI to characterize even more complex topological systems.

## 2  Model and entanglement witnesses

We study a generalization of the Kitaev chain [6] with an inhomogeneous (site-dependent) chemical potential $\mu_l$ and an algebraic decay of the superconductive pairing with exponent $\alpha$ [31, 85] :

$$\hat{H} = -\frac{J}{2}\sum_{l=1}^{L}\left(\hat{a}_l^\dagger \hat{a}_{l+1} + \hat{a}_{l+1}^\dagger \hat{a}_l\right) - \sum_{l=1}^{L}\mu_l \hat{a}_l^\dagger \hat{a}_l + \frac{\Delta}{2}\sum_{m=1}^{L}\sum_{l=1}^{L-1}d_l^{-\alpha}\left(\hat{a}_m \hat{a}_{m+l} + \hat{a}_{m+l}^\dagger \hat{a}_m^\dagger\right). \quad (1)$$

Here, $\hat{a}_l^\dagger$ ($\hat{a}_l$) is the creation (annihilation) operator of a spinless fermion at site $l$, $L$ is the number of lattice sites, $J > 0$ is the hopping amplitude, $\Delta$ is the pairing strength and $d_l = l$ for an open chain, while $d_l = \min(l, L - l)$ for a closed one. In the limit $\alpha \to \infty$, we have $d_l^{-\alpha} \to \delta_{l,1}$, where $\delta_{l,m}$ is the Dirac delta function, recovering nearest-neighbour pairing. Following Ref. [86], it is possible to rewrite the Hamiltonian Eq. (1) in a quadratic form and diagonalize the model exactly for any value of $\alpha$ and $\mu_j$, see footnote [87] for details.

The model (1) has been extensively studied in the literature, especially for a uniform chemical potential $\mu_l = \mu$ and in the limit $\alpha \to \infty$. A quantum phase transition at $|\mu|/J = 1$ separates a topological phase (for $|\mu|/J < 1$) from a topologically-trivial phase (for $|\mu|/J > 1$) [6]. When open boundary conditions are assumed, the topological phase is characterized by Majorana edge modes with anyonic statistics [6, 88], related to a $Z_2$ degeneracy of the ground state in the thermodynamic limit. The system belongs belongs to the $D$ class of topological insulators and superconductors, characterized by particle-hole symmetry $C$ [1, 2]. The topological phase and the related edge modes are topologically protected. Specifically, they are stable against perturbations (such as chemical potential inhomogeneities) that are weak compared to the mass gap and do not change the symmetry and the related symmetry-protected topology of their quantum state [3] The main effect of spatial inhomogeneities is to narrow the spectral gap of the lowest-energy excited state. Eventually, for sufficiently strong perturbations, the excitation gap closes and the Majorana modes no longer exist. The inhomogeneity-induced quantum phase transition associate to the fate of Majorana modes has been studied in Refs. [14–21] for nearest-neighbour coupling and different distributions of $\mu_l$.

Variable-range extensions ($\alpha < \infty$) of the Kitaev chain have also received relevant attention recently, especially in the case of a uniform chemical potential $\mu_l = \mu$ [31–33, 85, 89]. In this case, and for $1 \leq \alpha < \infty$, the system is known to be topologically equivalent to the SR Kitaev chain [32] with a topological Majorana phase for $|\mu|/J < 1$. For $\alpha < 1$, the situation changes completely. The corresponding diagram, for an homogenoeus chemical potential has been studied in Ref. [31, 33]. Purely LR insulating phases – that means not included in the classification of the SR topological insulators [90] – occur. A quantum phase transition is found at $\mu/J = 1$. The phase for $\mu/J < 1$ hosts massive edge modes (derived from the hybridization of the Majorana modes [85, 91], also indicated as Dirac modes), even in the thermodynamic limit, and a related LR topology [32] is realized. For $\mu/J > 1$ a LR phase, not hosting any edge mode, is present. It is interesting that, changing $\alpha$ across $\alpha = 1$, the transitions between SR and LR phases occur without closing the mass gap. We recall that the above behavior holds

for a uniform chemical potential. Less is known about the interplay of finite range pairing and chemical-potential inhomogeneities as well as the robustness of LR phases [25–27].

## Quantum Fisher information

The aim of this manuscript is to study the quantum Fisher information (QFI) of the ground state of Eq. (1). The QFI can be generally defined as the susceptibility of the quantum (Uhlmann's) fidelity $\mathcal{F}(\theta, d\theta)$ [92, 93] between a quantum state depending on a parameter $\theta$ and its infinitesimally-varied neighbour at $\theta + d\theta$ [94–96]. For pure states, as relevant for this work,

$$\mathcal{F}(\theta, d\theta) = |\langle \psi(\theta)|\psi(\theta + d\theta)\rangle| = 1 + \frac{(d\theta)^2}{4} F[|\psi(\theta)\rangle] + O(d\theta)^3, \tag{2}$$

where $F[|\psi(\theta)\rangle] = \frac{1}{2} d^2 \mathcal{F}/d\theta^2$ is the QFI. A convenient choice of parametric transformation is the unitary evolution $|\psi(\theta + d\theta)\rangle = e^{-i\hat{J}(\theta + d\theta)}|\psi\rangle$. Here, $\hat{J} = \sum_{l=1}^{L} \hat{j}_l$ is a collective operator, $l = 1, ... L$ labels $L$ subsystems and $\hat{j}_l$ is a Hermitian operator defined in the local Hilbert space $\mathcal{H}_l$ ($\mathcal{H} = \bigotimes_{l=1}^{L} \mathcal{H}_l$ being the total system's Hilbert space). This choice of transformation leads to

$$F[|\psi\rangle, \hat{J}] = 4 \sum_{l,m} \left( \langle \psi|\hat{j}_l \hat{j}_m|\psi\rangle - \langle \psi|\hat{j}_l|\psi\rangle \langle \psi|\hat{j}_m|\psi\rangle \right), \tag{3}$$

the QFI being the sum of the two-points connected correlations [67]. The QFI in Eq. (3) is an efficient witness of ME [57–59, 97]: the violation of the inequality $F[|\psi\rangle, \hat{J}]/(\Delta \hat{j})^2 \leq kL$ signals at least $k-$partite entanglement ($1 \leq k \leq L$) between the $L$ subsystems $\mathcal{H}_l$ [58, 59], where $(\Delta \hat{j})^2 \equiv (j_{\max} - j_{\min})^2$ is the difference between the maximum and the minimum eigenvalue of the local operator $\hat{j}_l$. Here we are assuming that all operators $\hat{j}_1, ..., \hat{j}_L$ have the same bounded spectrum. More in detail, separable states, e.g. pure states $|\psi\rangle_{\text{sep}} = \bigotimes_{l=1}^{L} |\psi_l\rangle$, satisfy $F[|\psi\rangle_{\text{sep}}, \hat{J}]/(\Delta \hat{j})^2 \leq L$ [57]. In other words, for separable states, the QFI scales, at best, extensively with the system's size. A super-extensive scaling, namely $F[|\psi\rangle, \hat{J}]/(\Delta \hat{j})^2 \sim L^\beta$ with $\beta > 1$ is only possible if $|\psi\rangle$ is multipartite entangled, with $k \sim L^{\beta-1}$. Notice that ME is not necessarily associated to a super-extensive scaling of the QFI. Finally, $F[|\psi\rangle, \hat{J}]/(\Delta \hat{j})^2 \sim L^2$ is the fastest possible scaling of the QFI (for the collective local operators considered here) and it is called the Heisenberg scaling [57]: it implies that $k \sim L$ [58, 59].

In the following, we study the optimized QFI,

$$F_{x,y}[|\psi_{\text{gs}}\rangle] = \max_{s} F[|\psi_{\text{gs}}\rangle, \hat{J}_{x,y}(s)], \tag{4}$$

where $|\psi_{\text{gs}}\rangle$ is the ground state of Eq. (1), and

$$\hat{J}_{x,y}(s) = \sum_{l=1}^{L} s_l \frac{\hat{\sigma}_{x,y}^{(l)}}{2}, \tag{5}$$

is a family of collective pseudo-spin operators with $s = \{s_1, ..., s_L\}$, $s_l = \pm 1$ being local sign coefficients. The Pauli operators $\hat{\sigma}_x^{(l)}$ and $\hat{\sigma}_y^{(l)}$ are expressed in terms of spinless fermionic operators via the Jordan-Wigner transformation [86, 98, 99]

$$\hat{\sigma}_{x,y}^{(l)} = \hat{a}_l^\dagger e^{i\pi \sum_{m=1}^{l-1} \hat{a}_m^\dagger \hat{a}_m} \pm e^{-i\pi \sum_{m=1}^{l-1} \hat{a}_m^\dagger \hat{a}_m} \hat{a}_l, \tag{6}$$

and are highly nonlocal in the fermionic lattice modes. In practice, taking into account that $\langle \psi_{\text{gs}}|\hat{\sigma}_{x,y}^{(l)} \hat{\sigma}_{x,y}^{(m)}|\psi_{\text{gs}}\rangle = 1$ and $\langle \psi_{\text{gs}}|\hat{\sigma}_{x,y}^{(l)}|\psi_{\text{gs}}\rangle = 0$ for all $l$, we have

$$F_{x,y}[|\psi_{\text{gs}}\rangle] = L + 2 \sum_{\substack{l,m=1 \\ l<m}}^{L} |\rho_{x,y}^{(l,m)}|, \tag{7}$$

where the correlation functions can be calculated following Ref. [86] as

$$\rho_x^{(l,m)} = \det \begin{pmatrix} G_{l,l+1} & G_{l,l+2} & \cdots & G_{l,m} \\ \vdots & \vdots & & \vdots \\ G_{m-1,l+1} & G_{m-1,l+2} & \cdots & G_{m-1,m} \end{pmatrix}, \tag{8}$$

$$\rho_y^{(l,m)} = \det \begin{pmatrix} G_{l+1,l} & G_{l+1,l+1} & \cdots & G_{l+1,m-1} \\ \vdots & \vdots & & \vdots \\ G_{m,l} & G_{m-1,l+2} & \cdots & G_{m,m-1} \end{pmatrix}, \tag{9}$$

and $G = -\Psi^T \Phi$, see footnote [87]. We are mostly interested in the scaling coefficients

$$\beta_{x,y} = \frac{d \log F_{x,y}[|\psi_{gs}\rangle]}{d \log L} \tag{10}$$

and, in particular,

$$\beta = \max\{\beta_x, \beta_y\}. \tag{11}$$

Multipartite entanglement witnessed by the QFI Eq. (4) does not depend on the basis of the multipartite decomposition but it depends on the choice of operator $J$ considered. In this respect, Eq. (5) is chosen as an educated guess, directly suggested by Jordan-Wigner mapping [33, 62].

Let us recall here previous results concerning the scaling of the QFI for the uniform chain $\mu_l = \mu$ and $\Delta \neq 0$ [33]. For SR coupling, $\alpha \geq 1$, it has been shown that, in the topological phase at $|\mu|/J < 1$, the scaling exponent in Eq. (11) is $\beta = 2$. In this phase, the QFI is maximized by the operator $\sum_{l=1}^{L} \hat{\sigma}_x^{(l)}/2$ [namely $s = \{1, 1, ..., 1\}$ in Eq. (5)]. This operator is the order parameter of the quantum Ising chain in a transverse field, corresponding, via Jordan-Wigner transformation, to the standard ($\alpha \to \infty$) Kitaev chain [98, 99]. In the following, we indicate this phase as xSR. Along the critical lines $\mu/J = \pm 1$, the QFI scales as $\beta = 3/2$, while in the trivial phase, $|\mu|/J > 1$, the scaling is $\beta = 1$. For LR coupling, namely $0 \leq \alpha < 1$, and $\mu/J < 1$, the QFI is still maximized by the operator $\sum_{l=1}^{L} \hat{\sigma}_x^{(l)}/2$ and has a scaling $\beta = 3/4$. We indicate this phase as xLR. Instead, for $\mu/J > 1$ the QFI is maximized by the operator $\hat{J}_y$ with staggered $s = \{-1, 1, -1, ...\}$, reaching $\beta = 3/4$. We indicate this phase as yLR. The xLR (yLR) phase is characterized by algebraically-decaying two-point correlations of the $\hat{\sigma}_x$ ($\hat{\sigma}_y$) operator and exponential decay of $\hat{\sigma}_y$ ($\hat{\sigma}_x$) correlations [33]. Along the critical line $\mu/J = 1$ we have again $\beta = 3/2$.

## 2.1 Topological invariant in the $\alpha \to \infty$ limit

In the limit $\alpha \to \infty$, we will compare the predictions of the QFI with the topological phases identified by a $Z_2$ topological index $\nu$. Assuming (anti-)periodic boundary conditions, $\nu$ can be calculated in various equivalent ways [100]. For instance, a common approach is the Berry phase [101]

$$\nu = \frac{i}{\pi} \int_{BZ} dk \, \langle u_k | \partial_k u_k \rangle, \tag{12}$$

where $|u_k\rangle$ is the positive (or, equivalently, the negative) eigenvector of $\hat{H}(k)$, the Fourier transform of Eq. (1), and the integral extends on the Brillouin zone. Assuming instead open boundary conditions, $\nu$ can be obtained from the transfer matrix equation [14–16]

$$\begin{pmatrix} \psi_{j+1} \\ \psi_j \end{pmatrix} = D_j \begin{pmatrix} \psi_j \\ \psi_{j-1} \end{pmatrix}, \qquad D_j = \begin{pmatrix} \frac{\mu_j}{\Delta+t} & \frac{\Delta-t}{\Delta+t} \\ 1 & 0 \end{pmatrix}, \tag{13}$$

where $\psi_i$ is the eigenfunction of the closest positive-energy eigenfunction above zero energy. In the topological phase, this is associated with the Majorana edge modes For a system of $L$ lattice sites, one calculates the matrix $\mathcal{D} = \prod_{i=1}^{L} D_i$. The two eigenvalues $\lambda_1$ and $\lambda_2$ of $\mathcal{D}$ fulfill the conditions $|\lambda_1 \lambda_2| < 1$. For $|\lambda_1| < |\lambda_2|$, one can define the topological invariant [14–16]

$$\nu = \frac{1 - \text{sgn}(\ln|\lambda_2|)}{2}. \tag{14}$$

This quantity assumes the values $\nu = 0$ in the trivial superconducting phase and $\nu = 1$ in the topological phase respectively. $\nu$ is related to the $Z_2$ parity number of the ground state and counts the number of Majorana modes by a direct manifestation of the bulk-boundary correspondence principle. We comment that, at least in homogeneous case $\mu_j = \mu$, the topological invariant in (12) and (14) can be equivalently calculated in terms of $\zeta = \text{Pf}[\hat{M}]/|\text{Pf}[\hat{M}]|$ [6, 102], where $\text{Pf}[\hat{M}]$ is the pfaffian of the Hamiltonian matrix $\hat{H}$, recast in a skew-symmetric form $\hat{M}$ by adopting a Majorana site-operator basis [100]. The pfaffian can change sign when the mass gap closes, as the various Hamiltonian parameter vary. Therefore, $\zeta$ can be taken as a $Z_2$ topological invariant (required by the $D$ symmetry of the Hamiltonian in Eq. (1) [1]), assuming values $\pm 1$. Physically, when open boundary conditions are assumed, $\zeta$ is also related with the $Z_2$ fermionic parity symmetry of the ground state of the Hamiltonian in Eq. (1), similarly as the $\nu$ index. Operatively, $\zeta$ can be evaluated via a closed formula for even-dimensional matrices, as in Ref. [103] . However, when perturbations breaking explicitly translational invariance are added, the pfaffian does not work in general as a topological index. This relevant situation will be described in Appendix, also for finite $\alpha$. Finally, other equivalent expressions for $\nu$ are known in the spatially unpertubed regime, in the infinite size-limit [104] as well as at finite sizes [105].

## 3   Commensurate quasiperiodic modulation (Harper potential)

We analyze here the following site-dependence of the chemical potential:

$$\mu_l = \mu + V \sin(2\pi l \omega + \phi), \tag{15}$$

where $\mu$ is a constant offset, $\omega = p/q$ is the commensurate modulation frequency, $p$ and $q$ are relative prime integer numbers, $V$ quantifies the strength of the inhomogeneities and $\phi \in [0, 2\pi)$ is a phase. The salient features of the topological phase diagrams discussed in this section show only a minor qualitative dependence on $\phi$.

For $\Delta = 0$, the Hamiltonian (1) with the Harper potential Eq. (15) and rational $\omega = p/q$ [106] maps exactly into a square lattice tight-binding model with constant magnetic flux $\cos(2\pi p/q)$ per plaquette (in units of $\hbar/e$, $e$ being the electric charge), where $V$ is the relative hopping along a second (cyclic) coordinate $m$ orthogonal to the linear direction $l$ in Eq. (1), and $\phi$ is the corresponding cyclic quantized momentum [107]. Depending on the ratio $p/q$ and on the filling, the present set-up hosts, beyond various metallic phases, many insulating phases, also with possible edge excitations [17, 107]. These phases occur following a characteristic fractal butterfly-like structure in the plane reporting the single-particle energies, as functions of the magnetic flux. If $\Delta \neq 0$, the same mapping as in Ref. [107] holds, with an additional pairing along the $l$ direction.

**Limit $\alpha \to \infty$**

The upper row panels of Fig. 1 show the scaling exponent $\beta$ of the QFI, Eq. (11), as a function of $\mu/J$ and $\omega$, and for different values of $V/J$. The bottom panels report the corresponding

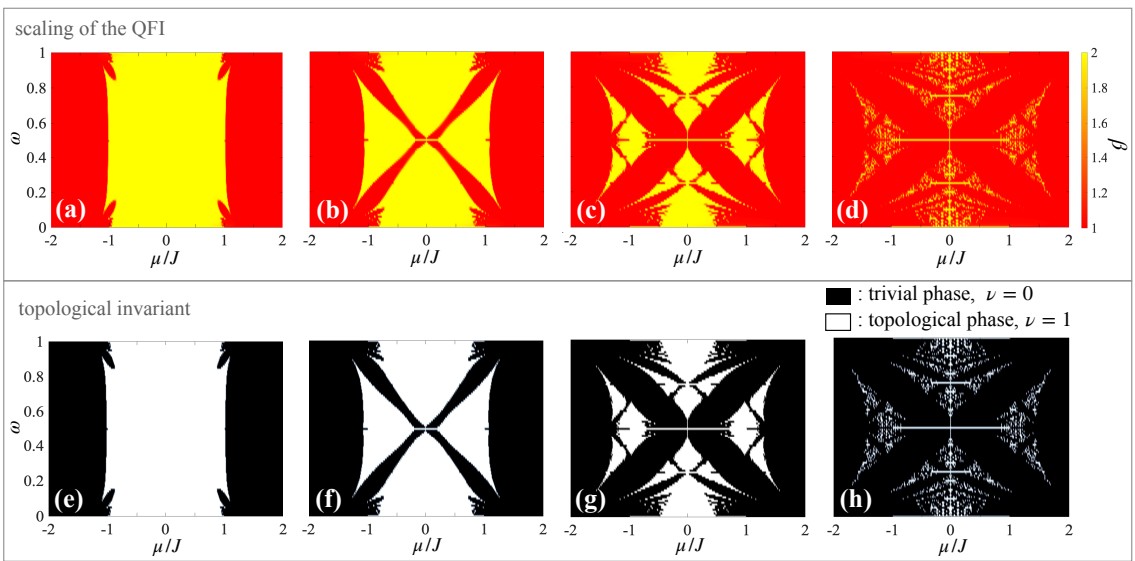

Figure 1: Phase diagram of the model (1) with nearest neighbour coupling ($\alpha = \infty$) and commensurate modulation frequency $\omega = p/q$, Eq. (15). Upper panels: scaling exponent $\beta$ of the QFI, defined in Eq. (11). Lower panels: topological invariant $\nu$, defined in Eq. (14). Different panels correspond to different values of $V/J$: 0.25 (a,e); 0.5 (b,f); 1 (c,g); and 1.25 (d,h). Here, $\Delta/J = 0.25$ and $\phi = 0$. The scaling $\beta$ is extracted from the calculation of the QFI between $L = 200$ and $L = 400$ with antiperiodic boundary conditions. Here $\omega$ is changed according to $p = 0, 1, 2, ..., 100$ and $q = 100$.

topological index $\nu$, calculated as in Eq. (14). We find a remarkable one-to-one correspondence between non-trivial topology, $\nu = 1$, and the Heisenberg-scaling, $\beta = 2$. Conversely, the trivial phase is characterized by an extensive scaling, $\beta = 1$. Recalling that $\phi = 0$ is assumed here, for $\omega = 0$, $1/2$ and $1$, the sinusoidal modulations in Eq. (15) vanish and we recover the homogeneous chemical potential case $\mu_l = \mu$. Accordingly, the topological phase is found for $|\mu|/J < 1$ [6,33], irrespective of $V$, as seen in Fig. 1. The robustness of Majorana edge states against the modulation strength $V$, for specific values of $\phi$ and $\omega$, has been also noticed in Ref. [108].

In the limit $V/J \to 0$ [e.g. $V/J = 0.25$ in panels (a) and (e)], we recover the non-trivial topological phase for $|\mu|/J \leq 1$ and any value of $\omega$. Increasing $V/J$, we find a typical Hofstadter-butterfly-like phase diagram. This characteristic structure was already noticed in Ref. [15] when studying the topological invariant $\nu$. For large values $V/J$ [e.g. $V/J = 1.25$ in panels (d) and (h)], the width of the topological phases becomes thinner and thinner, and eventually disappears for $V/J \gg 1$. It is worth pointing out that, in the phase diagrams of Fig. 1(a)-(d), the QFI is maximized by the operators $\sum_{l=1}^{L} \hat{\sigma}_x^{(l)}/2$ (namely $s = \{1, 1, ..., 1\}$ in Eq. (5) and $\beta = \beta_x = 2$] which is also the nonlocal order parameter for the case $V/J = 0$ [33]. Conversely, for all values of $\mu/J$, $\omega$ and $V/J$, we find $\beta_y = 1$.

Note that the rich butterfly structure shown in Fig. 1 disappears for $\Delta/J \geq 1$. In this case, one obtains a single (connected) topological region for $|\mu|/J \lesssim 1$, irrespective of $\omega$ and for every value of $V/J \lesssim 1$. Conversely, for $\Delta/J \geq 1$ and $V/J \gtrsim 1$, the phase diagram assumes a disperse multifractal structure.

### Finite-$\alpha$ regime

Figure 2 investigates how the butterfly-like phase diagram of Fig. 1 changes when decreasing $\alpha$. Figure 2(a) shows the scaling exponent $\beta$, as a function of $\mu/J$ and $\omega$, in the SR case $\alpha = 2$. Here we focus on the set of parameters $\Delta/J = 0.25$ and $V/J = 0.5$, which, in the the limit $\alpha \to \infty$, correspond to the phase diagram reported in Fig. 1(b). We find that the results obtained for $\alpha \to \infty$ hold qualitatively up $\alpha = 1$. The butterfly structure is essentially maintained for $\mu/J \geq 0$, while it is slightly distorted for negative values $\mu/J$. For $\alpha > 1$, the

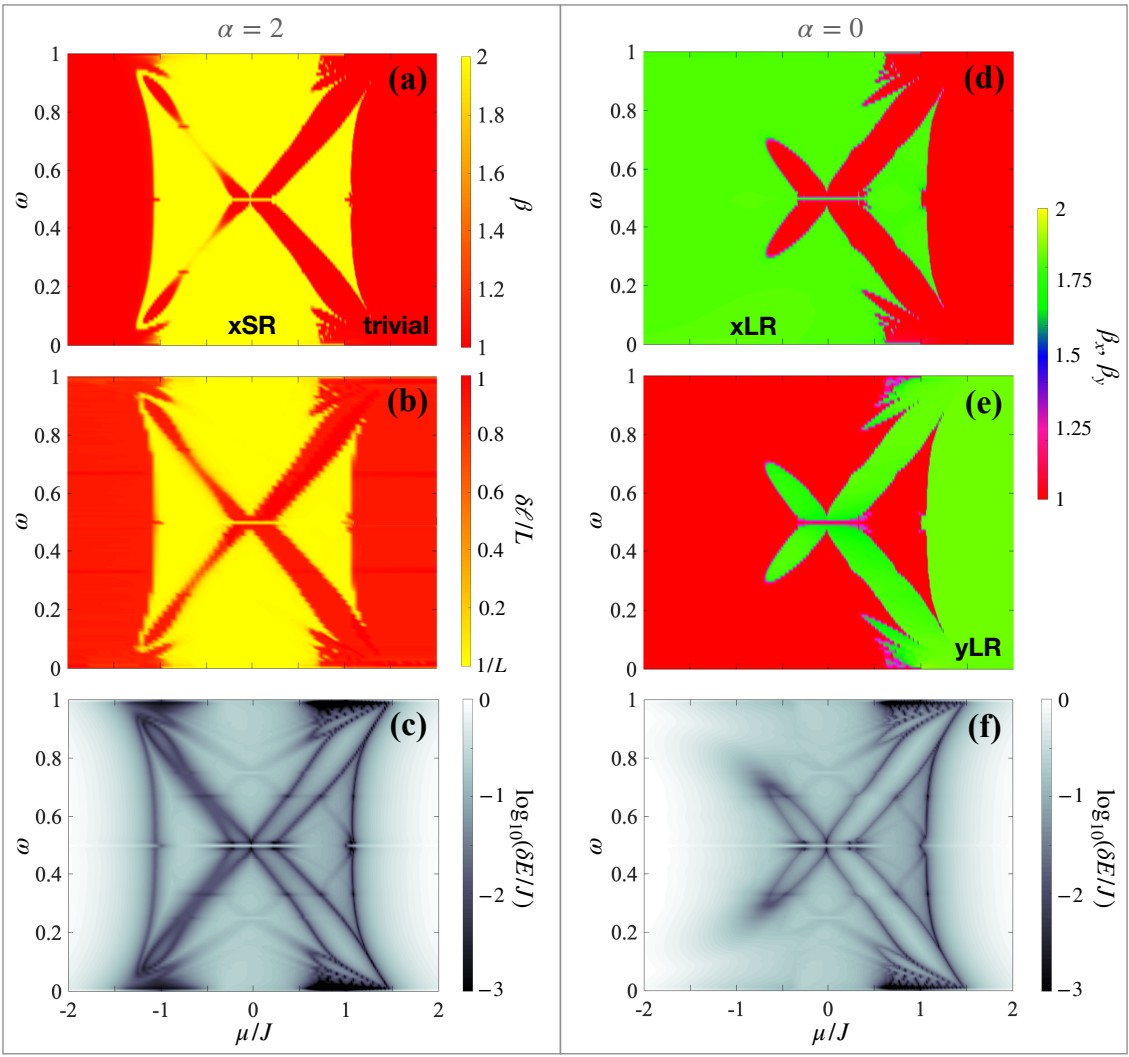

Figure 2: Phase diagram of the model (1) with finite-range coupling ($\alpha < \infty$) and commensurate modulation frequency, as a function of $\mu/J$ and $\omega = p/q$. Panel (a), (b) and (c) show the the scaling exponent $\beta$ of the QFI, the ELW $\delta\ell/L$ [see Eq. (16) and text], and the mass gap $\delta E/J$, respectively, in the SR case $\alpha = 2$. Panels (d) and (e) plot $\beta_x$ and $\beta_y$, respectively, in the case $\alpha = 0$. Panel (f) shows the corresponding mass gap. Here, $V/J = 0.5$, $\Delta/J = 0.25$ and $\phi = 0$. The scaling of the QFI is calculated between $L = 200$ and $L = 400$. The mass gap and the ELW are calculated for $L = 200$. The QFI and the mass gap are calculated in the closed chain with anti-periodic boundary conditions, while the ELW is calculated in the open chain. $\delta E/J$ is plotted with a lower cutoff at $10^{-3}$.

QFI is still maximized by the operator $\sum_{l=1}^{L} \hat{\sigma}_x^{(l)}/2$ and we identify a xSR phase when $\beta = 2$ [yellow region in Fig. 2(a)].

Probing the predictions of the QFI against a topological invariant is hindered by difficulties in calculating the later quantity for finite $\alpha$ and in the presence of chemical potential inhomogeneities (see Appendix). Instead, we have numerically checked that, in the open chain, the xSR phase is characterized by the presence of Majorana modes localized at the edges. To be more specific, in Fig. 2(b) we plot the edge localization width (ELW) defined as $\delta\ell = \ell_{\text{left}} + \ell_{\text{right}}$, where

$$\sum_{l=1}^{\ell_{\text{left}}} |\psi_l|^2 = C, \qquad \text{and} \qquad \sum_{l=L-\ell_{\text{right}}}^{L} |\psi_l|^2 = C. \qquad (16)$$

Here $\psi_l$ is the discrete normalized wavefunction of the first excited eigenstate (quasi-degenerate at finite size with the lowest-energy one in the Majorana phase). In other words, $\ell_{\text{left}}$ and $\ell_{\text{right}}$ count the number of lattice sites where $\psi_l$ is spatially localized with probability $C$ (in practice, we set $2C = 0.9$), starting from the left and right edges, respectively. In the presence of Majorana modes, we have $\ell_{\text{left}} \approx \ell_{\text{right}} \ll L/2$ (such that $\delta\ell/L \approx 1/L$), while for an extended wavefunction, $\ell_{\text{left}} \approx \ell_{\text{right}} \approx L/2$ (such that $\delta\ell/L \approx 1$). Furthermore, the borders of the xSR phase are characterized by the closing of the mass gap $\delta E/J$, as shown in Fig. 1(c),

Let us now turn to the case $\alpha < 1$. In Fig. 2(d) and (e), we plot $\beta_x$ and $\beta_y$, respectively. In each panel, the green region corresponds to a scaling coefficient $\beta_{x,y} = 3/4$, while the red region to $\beta_{x,y} = 1$. As we see, we still recover a asymmetric butterfly-like structure, which is qualitatively similar, especially for $\mu/J \geq 0$, to the one obtained for $\alpha > 1$. Following the uniform case, we thus identify xLR and yLR phases. Notably, the green and red regions in Fig. 2(b) and (c) complement each other perfectly, such that $\beta = \max(\beta_x, \beta_y) = 3/4$ for all value of $\mu/J$, except at the transition between xLR and yLR, where $\beta_x = \beta_y = 3/2$. This transition is marked by the closing of the mass gap, as shown in Fig. 1(f). We have also observed that the presence of massive Dirac modes is not a characteristic feature of the xLR phase: for instance, in the green region of 2(d) we observe localized as well as extended modes with no distinctive distribution. In other words, differently from Majorana modes for $\alpha > 1$, massive Dirac modes for $\alpha < 1$ are not robust against inhomogeneities.

Recently, Ref. [27] has reported the calculation of a real-space winding number (taken from the BDI class [109] and here indicates as $\tilde{\nu}$) for the inhomogeneous model considered here. To allow for a direct comparison with the QFI, we investigate, in Fig. 3, the phase diagram in the $\Delta/J$-$\alpha$ plane and for different values of $V/J$ (to be compared with Fig. 5 of Ref. [27]). The top panels show the scaling exponent $\beta$, while the bottom panels show the corresponding mass gap. The different column panels are obtained to different values of $V/J$: 0.5 (a,e); 1 (b,f); 1.5 (c,g) and 2 (d,h). The parameters considered here, $\mu = 0$, $\phi = \pi/2$ and $\omega = 72/117$ are analogous to the one in Ref. [27]. For relatively small values of $V/J$ and sufficiently large $\Delta/J$, we observe a xSR phase for relatively large $\alpha$ and a xLR phase for small $\alpha$. The transition between the two phases occurs without closing the energy gap, see Fig. 3(e). Instead, when increasing $V/J$, the phase diagram enriches substantially. A nested structure of lobes, each hosting a yLR phase, appears at sufficiently small values of $\alpha$ and $\Delta/J$. The lobes are separated by massless lines. Furthermore, the yLR lobe observed at small $\Delta/J$ is not separated by the trivial phases by a closing mass gap. Overall, our results concur well with those of Ref. [27]: the xSR region agrees with $\tilde{\nu} = 1$, while the xLR and yLR phases agree with $\tilde{\nu} = 0.5$ and $\tilde{\nu} = -0.5$, respectively. However, the study of the QFI addresses regions of the phase diagram where the calculation of $\tilde{\nu}$ in Ref. [27] is hindered by small values of the mass gap. This unveils a richer structure of phases than that shown in Ref. [27] and promotes the QFI as a relevant tool to study the phase diagram of topological systems, beyond the cumbersome analysis of topological invariants.

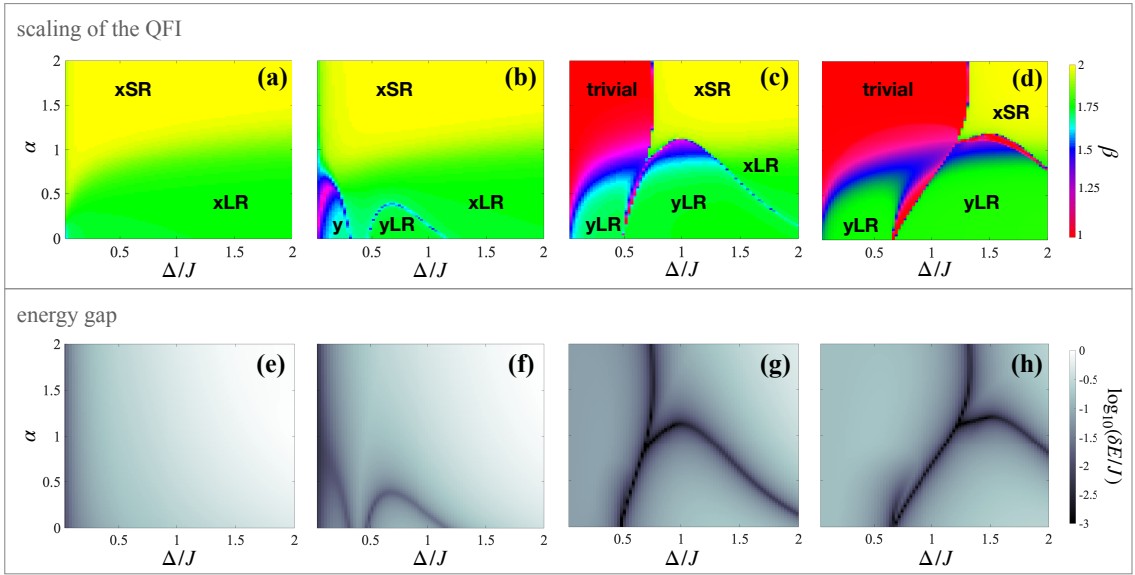

Figure 3: Phase diagram of the model in Eq. (1) with commensurate chemical potential, as a function of $\Delta/J$ and $\alpha$. Top panels: scaling exponent $\beta$. Bottom panels: energy mass gap $\delta E/J$ (plotted with a lower cutoff at $10^{-3}$). Different panels correspond to different values of $V/J$: 0.5 (a,e); 1 (b,f); 1.5 (c,g) and 2 (d,h). Here, $\mu/J = 0$, $\phi = \pi/2$ and $\omega = 72/117$. The scaling of the QFI is calculated between $L = 234$ and $L = 468$.

We finally comment that the structure of lobes shown in Fig. 3 can be linked to the butterfly structure observed in Figs. 1 and 2, as well to the fact that, also below $\alpha = 2$, only two phases are possible, not breaking (explicitly or spontaneously) the $Z_2$ parity and charge conjugation. Furthermore, the diagrams recovers, for large values of $\Delta/J$, the qualitative structure as in the limit $V \to 0$, dominated by the topological Majorana phase and by the xLR phases, as expected. The increase of $\Delta/J$ favours topology at every $\alpha$, while the increase of $V/J$ disadvantages it. This trend can be qualitatively understood considering the Hofstadter mapping discussed above [107]: the increase of $V/J$, that is mapped on the relative hopping along the second (orthogonal to $l$) direction, tends to disadvantage the effect of the $\Delta$ pairing, therefore destroying topology, for every $\alpha$.

## 4 Incommensurate quasiperiodic modulation (Aubry-Andre potential)

In this Section, we analyze the model in Eq. (1) as an onsite disorder is added. In particular, we consider the non-uniform chemical potential Eq. (15), where $\omega$ is now an *irrational* number, here set as the inverse golden ratio

$$\omega = \frac{\sqrt{5}-1}{2} . \tag{17}$$

In the case $\Delta = 0$, Eq. (1) reduces to the famous Aubry-André model [110, 111]. This model features a metal-insulator (localization-delocalization) transition, at the critical value $V_c/J = 1$, which can be determined exactly by a self-duality mapping [110]. For $V < V_c$, all eigenfunctions are extended, while for $V > V_c$ they are all localized. This transition has

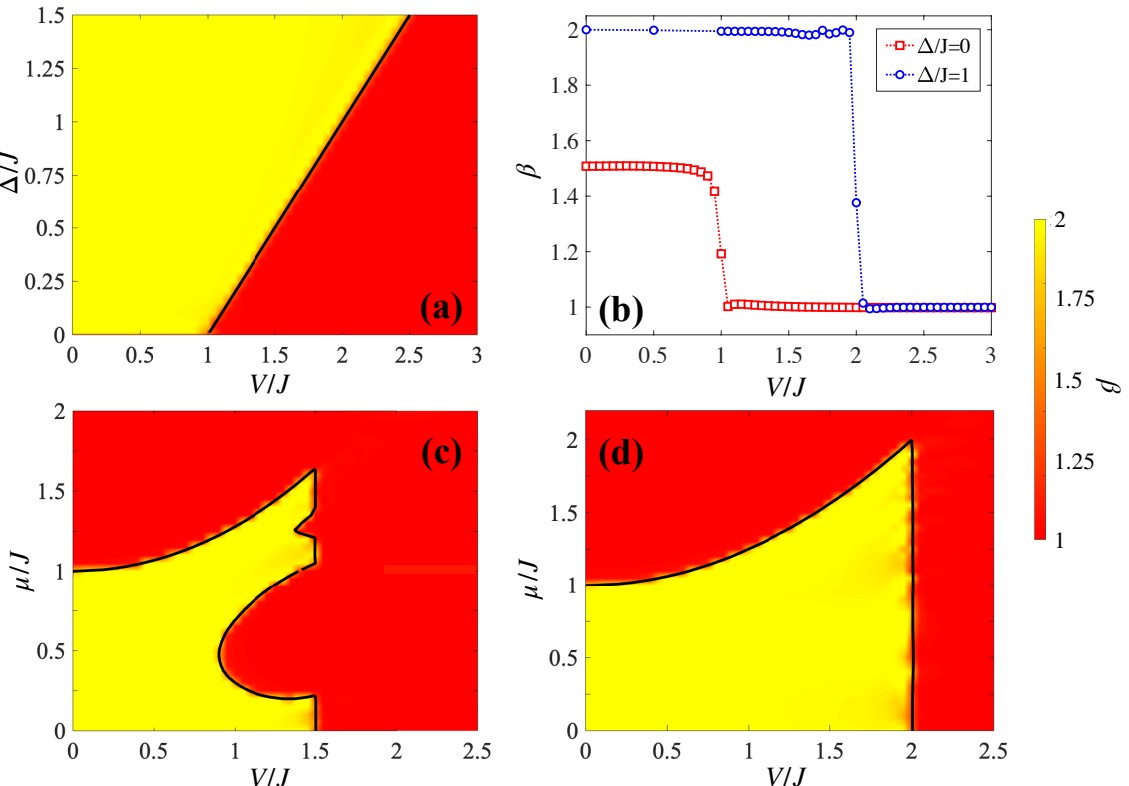

Figure 4: Phase diagram of the model (1) with SR coupling ($\alpha \to \infty$) and an incommensurate modulation frequency given by Eq. (17). (a) Power scaling $\beta$ of the QFI in the $V$-$\Delta$ plane, at $\mu/J = 0$. The black line is Eq. (18). (b) Cuts of the phase diagram in (a) for $\Delta/J = 0$ (red circles) and $\Delta/J = 1$ (blue circles). Panels (c) and (d) report the scaling coefficient $\beta$ in the $\mu - V$ plane for $\Delta/J = 0.5$, and $\Delta/J = 1$, respectively. The black line indicates the transition from a topological phase ($\nu = 1$) to a trivial one ($\nu = 0$), where $\nu$ is the $Z_2$ topological invariant calculated in Eq. (14). Here the QFI is calculated between $L = 100$ and $L = 200$ with open boundary conditions. The topological invariant is calculated for $L = 200$.

been observed experimentally with a non-interacting Bose-Einstein condensate trapped in a bichromatic lattice potential [112], see also Ref. [113], and with photons in optical waveguides [114, 115]. In the following, we set $\phi = \pi/2$ in Eq. (15) and study the system with open boundary conditions.

**Limit $\alpha \to \infty$**

For $\Delta \neq 0$, the model in Eq. (15) has been studied in Refs. [15, 18–20] in the limit $\alpha \to \infty$. For $\mu = 0$, a quantum phase transition takes place at a critical value of the disorder strength [15, 19]

$$\frac{|V_c|}{J} = 1 + \frac{\Delta}{J}. \tag{18}$$

When $V < V_c$, the $Z_2$ topological invariant (14) is $\nu = 1$ and the system thus hosts Majorana modes. Instead, for $V > V_c$, $\nu = 0$ holds and the system is in the topologically trivial superconductive phase.

Figure 4(a) shows $\beta$ in the $V$-$\Delta$ plane, for $\mu = 0$. The scaling of the QFI changes abruptly from super-extensive, $\beta = 2$, in the topological region ($V < V_c$), to extensive, $\beta = 1$, in the

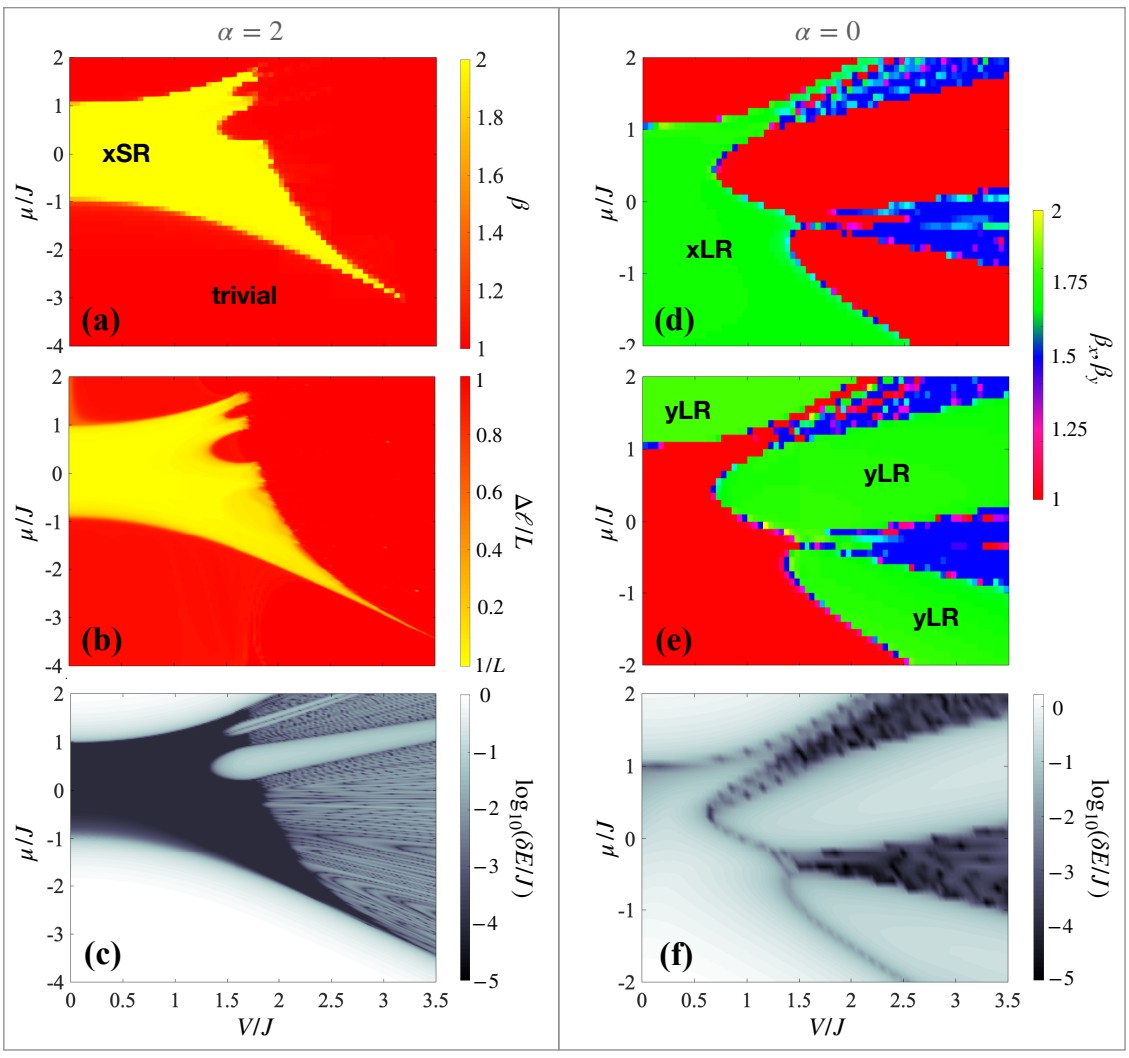

Figure 5: Phase diagram of the model in Eq. (1) with finite-range coupling ($\alpha < \infty$) and an incommensurate chemical potential frequency. Panels (a), (b) and (c) show, respectively, the scaling exponent $\beta$ of the QFI, the ELW and the mass gap, in the $\mu$-$V/J$ plane and for $\alpha = 2$. Panels (d), (e) and (f) corresponds instead to the case $\alpha = 0$. We plot $\beta_x$ and $\beta_y$, in panels (d) and (e) respectively, and the mass gap in panel (f). In all panels $\Delta/J = 1$.

trivial superconductive region ($V > V_c$). The sharp change occurs exactly along the critical line Eq. (18). Figure 4(b) shows cuts of the phase diagram of panel (a) for $\Delta/J = 0$ (red squares) and $\Delta/J = 1$ (blues circles). The metallic phase at $\Delta/J = 0$ and $V/J = 0$ is characterized by $\beta = 3/2$, which is intermediate between the topological Majorana phase and the trivial one. Then, the Aubri-Andrè localization transition, again at $\Delta/J = 0$, is signaled by the abrupt change of scaling of the QFI, jumping to $\beta = 1$ at $V/J = 1$, in agreement with Ref. [110].

We further study the case $\mu \neq 0$. The lower panels of Fig. 4 show the phase diagram for the scaling parameter $\beta$ in the $V$-$\mu$ plane, for $\Delta/J = 0.5$ (c) and $\Delta/J = 1$ (d). We clarify here that the phase diagrams of Fig. 4(c) and (d) are mirror-symmetric for $\mu < 0$. For $|\mu|/J \leq 1$, disorder competes with the Majorana phase, that disappears for sufficiently large value of $V$. The critical value of $V$ signaling the quantum phase transition point is essentially the same as $\mu/J = 0$ when $\Delta/J \geq 1$, while for $\Delta/J < 1$ there are characteristic re-entrances, already noticed in

Ref. [20]. For $|\mu|/J > 1$, disorder may even enlarge the topological phase: for instance in Fig. 4(d), e.g. at $\mu/J = 1.5$, we see that the phase is topologically trivial for $0 \leq V/J \lesssim 1.5$, while it becomes nontrivial for a sufficiently large disorder, specifically for $1.5 \lesssim V/J < 2$. The disorder-induced quantum phase transition predicted by the QFI is found in perfect agreement with the one obtained by calculating the $Z_2$ topological invariant [15, 19, 20], shown by the black line in Figs. 4(c) and (d).

**Finite-$\alpha$ regime**

Figure 5 clarifies how the $\mu$-$V$ phase diagram changes with $\alpha$. Here $\Delta/J = 1$, therefore, the corresponding diagram in the limit $\alpha \to \infty$ is the one reported in Fig. 4(d). In Fig. 5(a) we report the case $\alpha = 2$. Although the diagram still shows the characteristic transition from $\beta = 2$ (xSR) to $\beta = 1$ (in the trivial superconductive phase), we find qualitative changes with respect to the SR case. First, the diagram is not symmetric any longer with the respect to the change of sign of $\mu$. Furthermore, for $\mu > 0$, there appear again characteristic re-entrances, similarly as discussed in the previous subsection. We further compare the prediction of the QFI with the ELW, as calculated in Eq. (16), see Fig. 5(b), and the mass gap, see Fig. 5(c). Overall, we find a perfect agreement between the xSR region and that hosting Majorana edge modes. As shown in Fig. 5(c), the characteristic structures of the phase diagram prolong, for $V$ larger than a critical value, into regions where the mass gap is small but finite: as a revealed by a finite-size scaling that shows a saturation of $\delta E$ as a function of the system's size. Instead, the characteristic re-entrances prolong into regions of much larger mass gap. In Ref. [20] (for the case $\alpha \to \infty$), the latter feature was associated to a band gap region, while the former to an Anderson insulator regime. The scaling of the QFI does not reveal the difference between these two different kind of regions since both correspond to a trivial topological phase.

For $\alpha < 1$ the diagram changes abruptly. In Fig. 5(d) and (e), we plot $\beta_x$ and $\beta_y$, respectively, for $\alpha = 0$. The different, xLR and the yRL regions (with $\beta_{x,y} = 3/4$, respectively) complement each others, except in extended regions where $\beta = 3/2$ (which appears in blue in the figures) that are associated to a small mass gap, see Fig. 5 (f). It should be noticed that $\beta = 3/2$ is also found along the line of vanishing energy gap that separates the xLR and the yRL regions. In the same panels (d) and (e), extended quasi-critical regions, denoted in blue, are observed. The analysis of the mass gap in the quasi-critical regions is unclear: the gap is typically small but a finite-size analysis shows a clear decrease for some values of the parameters and a saturation for nearby values. No localized Dirac modes are observed for parameter values in this region.

## 5 Uncorrelated random disorder (Anderson potential)

In this Section, we consider a genuine random disorder *á la Anderson* [116], that means

$$\mu_l = \mu + \tilde{\mu}_l, \tag{19}$$

with $\tilde{\mu}_l \in [-V, V]$ distributed randomly. Again, we assume closed (antiperiodic) boundary conditions. For $\Delta/J = 0$, a perturbatively small amount of disorder is sufficient to induce localization in one dimension, for a sufficiently large system [117, 118].

**Limit $\alpha \to \infty$**

This case is qualitatively similar to the incommensurate quasiperiodic case discussed in the previous section: by varying $V/J$, we observe a disorder-induced quantum phase transition from a Majorana phase, with $\bar{\nu} = 1$, to a superconducting trivial one, where $\bar{\nu} = 0$. Here, we

indicate as $\bar{\nu}$ as the average of the $Z_2$ topological invariant Eq. (14) over independent realizations of the disorder. The disorder-induced quantum phase transition has been characterized by probing the disappearance of the Majorana modes [15, 19, 21], also studying the decay of the entanglement entropy, as well as the degeneracies of the entanglement spectrum [54]. Instead, in Ref. [23], the systems has been studied by looking at the presence of LR order in the correlation functions of the operator $\sigma_x^{(l)}$.

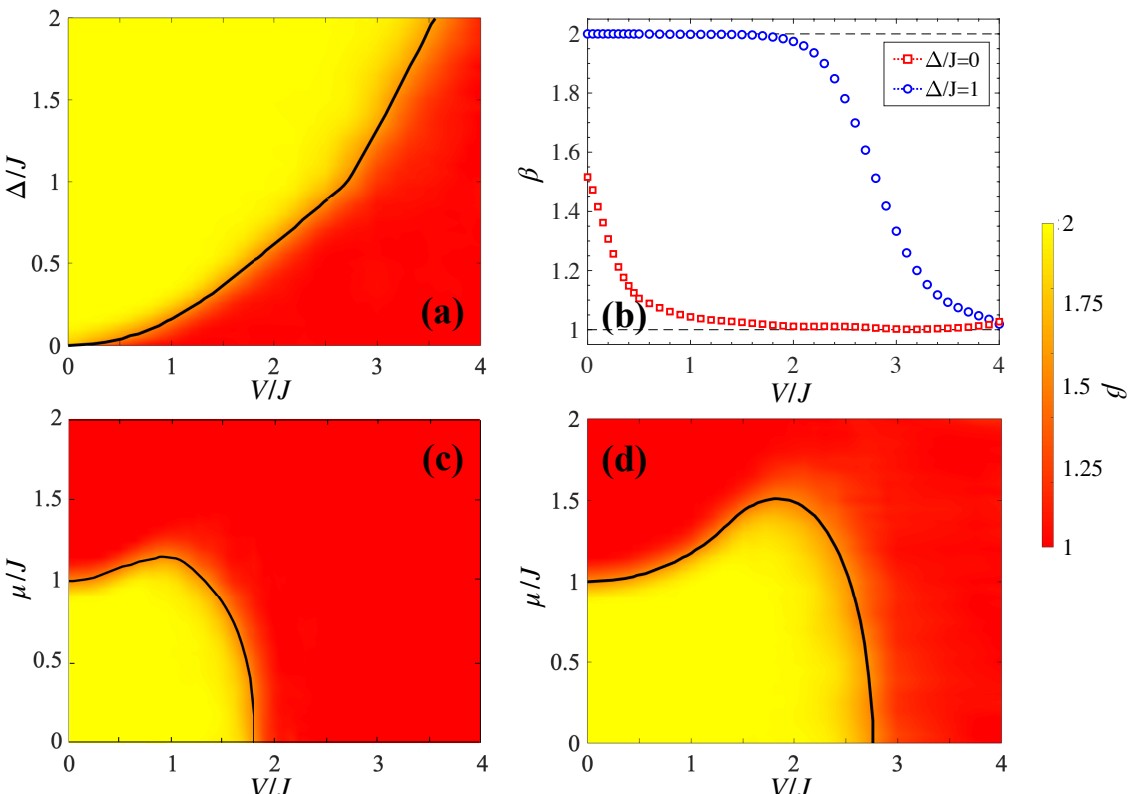

Figure 6: Phase diagram of the model (1) with SR coupling ($\alpha \to \infty$) and and Anderson disorder, Eq. (19). (a) $\beta$ in the $V$-$\Delta$ plane, at $\mu/J = 0$. The black line is $\bar{\nu} = 0.5$, where $\bar{\nu}$ is the disordered-averaged topological invariant. (b) Cuts of the phase diagram in (a) for $\Delta/J = 0$ (red squares) and $\Delta/J = 1$ (blue circles). Panels (c) and (d) show $\beta$ in the $V$-$\mu$ plane, for $\Delta/J = 0.5$ and 1, respectively. The black line is $\bar{\nu} = 0.5$. Notice that the obtained phase diagram is symmetric under the transformation $V \to -V$ and $\mu \to -\mu$ (therefore, here we show the first quadrant only). Here the QFI is calculated between $L = 100$ and $L = 200$ with open boundary conditions. The topological invariant is calculated for $L = 200$. Averaging is performed over 500 disorder realizations.

Here, we calculate the QFI of the ground state of Eq. (1) for several independent realizations of the disordered potential in Eq. (19). We then calculate the disorder-average QFI and extract the scaling coefficient $\beta$ by varying the system's size. Figure 6 reports the obtained results. In the panel (a), we show the scaling $\beta$ in the $V$-$\Delta$ plane and for $\mu/J = 0$. Similarly as in Fig. 4, we observe a transition from a regime where the scaling $\beta = 2$ is not affected by the disorder, to a region at larger $V$, where $\beta = 1$. The black line in the figure indicates the values of $\mu/J$ and $V/J$ for which $\bar{\nu} = 0.5$ [15]: it separates sharply a topological phase with $\bar{\nu} = 1$ (for small $V/J$) from a trivial phase with $\bar{\nu} = 0$. In particular, at $\Delta/J = 1$, the QPT occurs at $V/J = e$ [15]. In Fig. 6(b), we show $\beta$ as a function of $V/J$, for $\Delta/J = 0$ (red squares) and

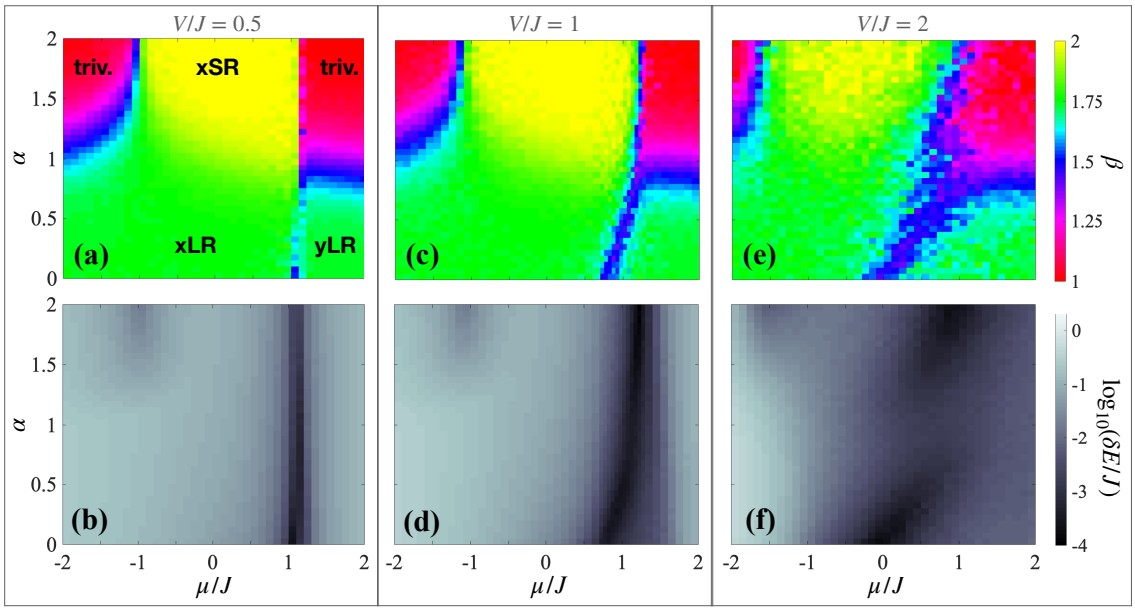

Figure 7: The upper row panels show $\beta$, as a function of $\mu/J$ and $\alpha$, and for different values of $V/J$: 0.5 (a), 1 (b) and 2 (c). The bottom panels show the corresponding (disorder-averaged) mass gap. Here $\Delta/J = 1$ and other parameters are as in Fig. 6.

$\Delta/J = 1$ (blue circles). For $\Delta/J = 0$, the metallic phase is not robust against disorder and the scaling coefficient $\beta$ decays quickly from $\beta = 1.5$ (at $V = 0$) to $\beta = 1$. In general, for $\Delta \neq 0$, the transition between $\beta = 2$ and $\beta = 0$ is much smoother than the one observed in Fig. 4 due to disorder fluctuations.

Finally, in Fig. 6(c) and (d), we plot the exponent $\beta$ in the $V$-$\mu$ plane, for $\Delta/J = 0.5$ (c) and $\Delta/J = 1$ (d), respectively. Again, the transition from the Majorana phase to the superconductive trivial one (along the black line identified by $\bar{\nu} = 0.5$) is marked by a change of the scaling coefficient $\beta$. The phase diagram is qualitatively similar to the one obtained for the incommensurate Aubry-Andre potential in Fig. 4. In particular, disorder enlarges the Majorana phase [23, 54]: this effect is due to the fact that here disorder is such that $\mu_l$ in Eq. (19) is symmetrically distributed around $\mu$. Therefore, starting for instance from the not topological phase and close to the phase transition, for small-enough $V$, disorder selects some Hamiltonian configurations (which we mediate on) lying in the topological phase [119, 120], eventually favouring it.

**Finite-$\alpha$ regime**

For Anderson disorder, the phase diagram in the $V$-$\mu$ plane is very similar as in the incommensurate case of Fig. 5 and we do not show it explicitly here. Instead, in the various panels of Fig. 7, we plot $\beta$ in the $\mu/J$-$\alpha$ plane, for $\Delta/J = 1$ and different values of $V/J$: 0.5 (a), 1 (b) and 2 (c). The corresponding diagram for $V = 0$ (namely $\mu_l = \mu$) has been studied in Ref. [33]. The phase diagram for sufficiently small values of $V$ [e.g. $V/J = 0.5$ in panel (a)] shows features very similar to the case $V = 0$. In particular, we recognize a xSR phase for $\alpha > 1$ and $|\mu|/J \leq 1$, a xLR phase for $\mu/J \lesssim 1$ and $\alpha < 1$, where $\beta = \beta_x = 3/4$, and a yLR phase for $\mu/J \gtrsim 1$ and $\alpha < 1$, where $\beta = \beta_y = 3/4$. The transition between the xLR and yLR phases, as well as between the xSR and the trivial phases, is marked by a closing mass gap around $\mu/J = 1$, where $\beta = 3/2$. Increasing $V$, the critical region shifts and enlarges following closely the behaviour of the mass gap, see Figs. 7(b), (d) and (f). The topological xSR shrinks

while the xLR and yLR phases appear quite robust against disorder, especially for large values of $|\mu|/J$.

## 6  Discussion

Finally, it is interesting to make a direct comparison between commensurate, incommensurate and Anderson inhomogeneities. In the upper panels of Fig. 8 we plot $\beta$ in the $V/J$-$\alpha$ plane. The commensurate case is calculated for $\omega = 72/117 \approx 0.61$, which is very close to Eq. (17) for the incomemnsurate case. The phase diagrams are similar in the three cases. Yet, the yLR phase found in the commensurate case for large values of $V$ and small $\alpha$ is strongly affected by disorder, turning into an extended "critical region" with $\beta = 3/2$ and a small mass gap. Furthermore, the lobe structure shown for commensurate and incommensurate potentials for relatively small values of $V/J$ and $\alpha$ is washed out for Anderson disorder. The border between the xSR and the trivial regions, as well as between the xLR and yLR regions, agree well with the closing of the mass gap, as shown in bottom panels of Fig. 8

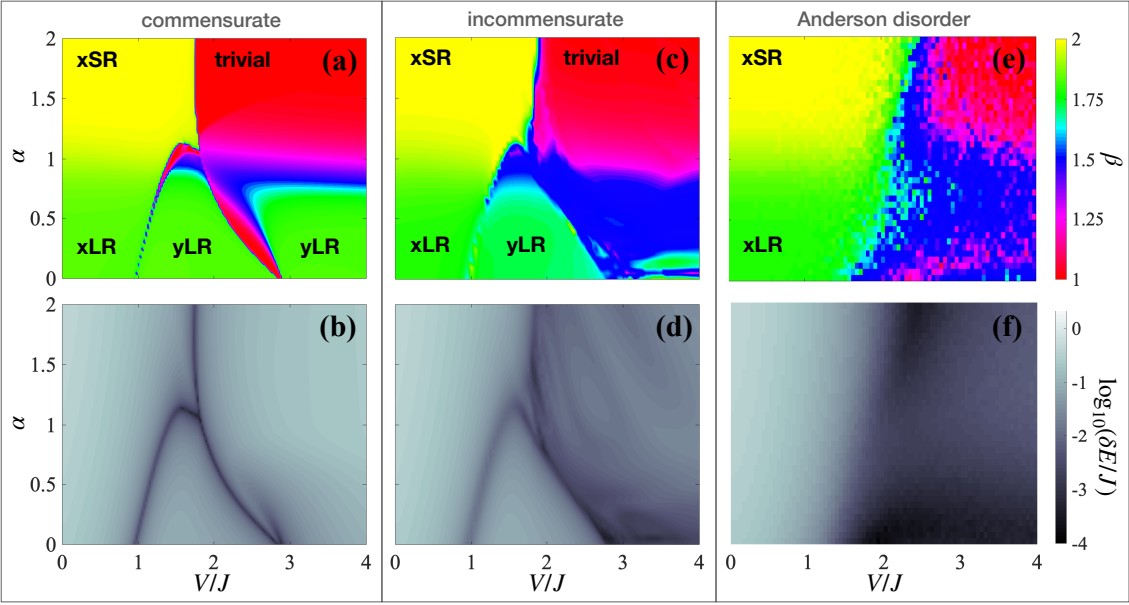

Figure 8: Phase diagrams showing the scaling of the QFI in the $V/J$-$\alpha$ plane for commensurate (a), incommensurate bichromatic (b) and Anderson (c) potentials for $\mu = 0$. The lower panels show the corresponding mass gap. The commensurate bichromatic case is calculated for $\omega = 72/117$ and anti-periodic boundary conditions. In this case, the scaling coefficient $\beta$ is calculated within $L = 234$ and $L = 468$, while the mass gap for $L = 234$. The incommensurate bichromatic case is shown for $\omega$ given by Eq. (17) and open boundary conditions. In this case, the scaling coefficient $\beta$ is calculated within $L = 100$ and $L = 200$, while the mass gap for $L = 200$. For the disordered Anderson potential case, $\beta$ is calculated within $L = 100$ and $L = 200$ and anti-periodic boundary conditions, while the mass gap is calculated for $L = 200$. In this case, we considered averaging over 500 realizations of the disorder. In all panels, we set $\Delta/J = 1$.

# 7   Conclusions

To summarize, in this work, we have discussed the stability of the ME against inhomogenities in paradigmatic gapped fermionic wires, also hosting symmetry-protected topological phases. In particular, we have calculated the QFI of the ground state of a generalization of the celebrated Kitaev chain with variable-range pairing and in the presence of periodic, quasiperiodic, as well as genuinely random offsets. Our analysis conveys two key messages. First, the scaling of the QFI allows to identify clearly topological and long-range regions. Conversely, topological invariants are difficult to define and/or calculate in the presence of inhomogeneities and long-range pairing. Similarly, the analysis of correlation functions is notoriously difficult for inhomogeneous systems, as they can show complicated staggered structure, while the QFI is characterized by clear power-law scalings. In particular, in the limit $\alpha \to \infty$, the topological regions identified by the QFI agrees perfectly with that singled out by the $Z_2$ topological invariant [14–16]. For $\alpha < \infty$ the border of the different phases agree well with the behaviour of the mass gap when the latter is relevant to determine a transition between trivial and topological phases. In the LR case, our methods go beyond the current literature [20, 27], where a characterisation of the phase diagram was hindered by difficulties associated to the identification of topological invariants. Second, the ME identified by the QFI appears robust against inhomogeneities. In some cases, the QFI is even favoured by the disorder. In particular, the super-extensive scaling of the QFI is as robust as the non-trivial phases themselves: the stability of ME holds at least until (SR and LR) topologies are preserved by the Hamiltonian symmetries, not broken by the added inhomogeneities, and by the mass gap. This work promotes the QFI as a valuable tool to analyze systems hosting symmetry-protected topological phases and opens toward the study of even more interesting topological systems, even in the presence of inhomogeneities and disorder. The study of the QFI presented here can be also extended to analyze the dynamical evolution of entanglement, e.g. under braiding [121].

## Acknowledgements

We are thank M. Burrello, A. Nava, and A. Smerzi for discussions.

**Funding information**   We also acknowledge financial support from the European Union's Horizon 2020 research and innovation programme - Qombs Project, FET Flagship on Quantum Technologies grant no. 820419. L. L. also acknowledges research funding from the PRIN project 2017 number 20177SL7HC, financed by the Italian Ministry of education and research.

# 8   Appendix

Along the main text, the calculation of the QFI is sometimes compared to the calculation of topological invariants, as in Section 2, signaling different quantum phases. For this reason, in this Appendix, we recall them further, also for finite $\alpha$. Importantly, we also discuss important limitations of these indices occurring for some cases that we have dealt with.

If $1 < \alpha < \infty$, the topological index $\nu$ can be evaluated both as in Eq. (12), if antiperiodic conditions are assumed, and via the pfaffian. Instead, the transfer matrix method, leading to Eq. (14), looks quite cumbersome.

Instead, if $\alpha < 1$, the topological index cannot be calculated as meantioned in the main text and above. There, a counterpart of Eq. (12), can be calculated by assuming close boundary

conditions [32]:

$$\frac{1}{2} + \frac{i}{\pi} \int_{BZ} \mathrm{d}k \, \langle u_k | \partial_k u_k \rangle = \frac{1}{2} + \nu. \tag{20}$$

This assumes the value 1 if $\mu/J < 1$, counting the massive edge modes, while it is equal to 0 if $\mu/J > 1$, where no protected edge states are present at all. Instead, $\nu$, Eq. (12), is badly defined for $\alpha < 1$, assuming semi-integer values [27, 32]. Notably, the critical line $\alpha = 1$, although locating a beyond-first-order phase transition, is characterized by a nonvanishing mass gap in the thermodynamic limit. Consequently, the $Z_2$ parity number symmetry is restored, also with open boundary conditions, as also clear in the structure of the entanglement spectrum [32]. All these fact are direct consequences of the algebraic tails of the two-points correlations of the model, still in massive regimes. Finally, around the critical and massless line $\mu/J = 1$, conformal invariance breaks down below $\alpha = 1$. Most of the described features can be linked with the dynamics of states, called "singular", at the border of the Brillouin zone of the model, with diverging energy [89].

Furthermore, a suitable quantity, distinguishing the LR phases among each others, as well as from the SR one, is still the pfaffian. To this aim, let us recall that, in momentum space, the sign of the pfaffian can be reduced to [6]

$$\zeta = \mathrm{sign}\Big[ \mathrm{Pf}\big(\hat{M}(k=0)\big) \mathrm{Pf}\big(\hat{M}(k=\pi)\big) \Big]. \tag{21}$$

Indeed, the charge conjugation proper of the $D$ class implies that mass gap of the Hamiltonian operator, $\hat{H}(k)$, in momentum space and of the corresponding antisymmetric operator $\hat{M}(k)$, can vanish at opposite sign pairs of momenta $\pm k$. However, these pairs does not contribute to the change of sign on the pfaffian. Instead, the change of sign can occur at the momenta $k = 0, \pi$, auto-conjugated under $C$. These momenta are precisely those where the mass-gap closes at the quantum transition points for the LR extensions of Kitaev chain in Eq. (1). We notice that the pfaffian is ill-defined at $k = 0$, where $\hat{M}(k=0)$ ($\hat{H}(k=0)$) becomes singular if $\alpha < 1$, and $\lim_{k\to 0^-} \hat{M}(k) \neq \lim_{k\to 0^+} \hat{M}(k)$ (similarly as for $\hat{H}(k)$). Such a type of is called a "second-type singularity" in Ref. [32], and leads to long-range phases, contrary to the "first-type singularities", where the two limits above coincide. Instead, the long-range phase are still distinguished by the change of sign $\mathrm{Pf}\big(\hat{M}(k=\pi)\big)$, also assuming values $\tilde{\zeta} = \pm 1$, as the various Hamiltonian parameter vary. ffFWe notice finally that the pfaffian is not related to the fermionic number parity of the ground-state, here always positive, due to the fact that the edge modes are massive. Therefore, although still useful, as it changes sign at a second-order phase transition, it is not a genuine topological invariant, being not related to the number of edge modes. However, here the possibility of a change of $\zeta$ without any change of topology (then say between topologically trivial long-range phases) is eventually ruled out in the absence of other mechanism inducing quantum phase transitions, as the breakdown (explicit or spontaneous) of the $Z_2$ symmetry oh the Hamiltonian in Eq. (1), or the introduction of further singularities, as due to other long-range Hamiltonian couplings in Eq. (1).

When spatial inhomogeneities are added, the sign of the pfaffian is unreliabile to signal SR ans/or LR phases. We encountered the described situations for instance in Fig. (2): there the pfaffian can rapidly oscillate, also in the absence of energy gap closings, or not to change where closing arise. Indeed, in presence of inhomogeneities, the pfaffian, evaluated as in can be extremely small (compared with the typical Hamiltonian energy scale, $J$ in our case) in modulus in certain ranges of the Hamiltonian parameters, and already at limited space sizes. Therefor, due to the limited numerical precision, the sign of the pfaffian can be not sufficiently stable, instead oscillating also in the absence of energy gap closings. Moreover, mass gap closings, related with certain quantum phase transitions, can involve pairs of degenerate lowest-energy

levels (as pairs of momenta different from $k = (0, \pi)$, connected by the charge-conjugation symmetry [6]). In this condition, the pfaffian does not change as a similar transition occurs, also between different topologies.

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

Explicitly, for open boundary conditions, they read $A_{ij} = -\mu_j \delta_{i,j} - \frac{J}{2}(\delta_{i,j+1} + \delta_{i,j-1})$, and

$$B_{ij} = \frac{\Delta}{2} \begin{cases} \frac{1}{|i-j|^\alpha} & \text{if } i > j, \\ 0 & \text{if } i = j, \\ -\frac{1}{|i-j|^\alpha} & \text{if } i < j. \end{cases} \tag{22}$$

In the case of anti-periodic boundary conditions, the same expression hold, with $\hat{a}_{j+L} = -\hat{a}_j$. These models can be diagonalized exactly via the singular value decomposition, $\mathbf{\Phi}^T \mathbf{\Lambda} \mathbf{\Psi} = \mathbf{A} + \mathbf{B}$, where $\mathbf{\Lambda}$ is the diagonal matrix of eigenvalues, $\mathbf{\Phi}$ and $\mathbf{\Psi}$ are $L \times L$ normalized matrices ($\mathbf{\Phi}\mathbf{\Phi}^T = \mathbf{\Psi}\mathbf{\Psi}^T = 1$) [86]. The corresponding eigenvectors read:

$$\eta_n = \sum_j g_{nj} a_j + h_{nj} a_j^\dagger, \qquad 1 \le n \le L. \tag{23}$$

The matrix elements $\phi_{nj} = g_{nj} + h_{nj}$ and $\psi_{nj} = g_{nj} - h_{nj}$ are the wavefunctions of the two Majorana modes $(\eta_n + \eta_n^\dagger)/2$ and $i(\eta_n - \eta_n^\dagger)/2$, with energy $\Lambda_n$, while $g_{nj}$ and $h_{nj}$ are the wavefunctions of $\eta_n$ and $\eta_n^\dagger$.

Exploiting the $\mathbf{A}$ and $\mathbf{B}$ parametrization of $\hat{H}$, the pfaffian of the corresponding skew-symmetric matrix $\hat{M}$ can be calculated as

$$\text{Pf}[\hat{M}] = (-1)^{(L(L-1)/2)} \text{Det}[\mathbf{A} + \mathbf{B}], \tag{24}$$

where $L$ is the number of lattice sites in Eq. (1).

[88] A. Stern, *Anyons and the quantum Hall effect—A pedagogical review*, Ann. Phys. **323**, 1 204-249 (2008).

[89] L. Lepori, D. Vodola, G. Pupillo, G. Gori, and A. Trombettoni, *Effective theories and breakdown of conformal symmetry in an long-range quantum chain*, Ann. Phys. **374** 35-66 (2016).

[90] A. Altland and M. R. Zirnbauer, *Nonstandard symmetry classes in mesoscopic normal-superconducting hybrid structures*, Phys. Rev. B **55**, 1142 (1997).

[91] K. Patrick, T. Neupert, and J. K. Pachos, *Topological Quantum Liquids with Long-Range Couplings*, Phys. Rev. Lett. **118**, 267002 (2017).

[92] A. Uhlmann, *The "transition probability" in the state space of a\*-algebra*, Rep. Math. Phys. **9**, 273 (1976).

[93] R. Jozsa, *Fidelity for mixed quantum states*, J. Mod. Opt. **41**, 2315 (1994).

[94] S. L. Braunstein,. and C. M. Caves, *Statistical distance and the geometry of quantum states*, Phys. Rev. Lett. **72**, 3439 (1994).

[95] L. Pezzè, Y. Li, W. Li, and A. Smerzi, *Witnessing entanglement without entanglement witness operators*, PNAS **113**, 11459 (2016).

[96] J. S. Sidhu and P. Kok, Geometric perspective on quantum parameter estimation, AVS Quantum Sci. **2**, 014701 (2020).

[97] We recall that a pure state $|\psi\rangle$ is $k$-partite entangled if it can be written as $\otimes_n |\psi_n\rangle$, where $|\psi_n\rangle$ is a joint state of $L_n \le k$ parties ($\sum_n L_n = L$ is the total number of parties in the system) that does not factorize. In other words, $k$-partite entanglement indicates the number of parties in largest non-separable subset [58, 59, 122]. A mixed state is $k$-partite entangled if it can be written as a convex combination of $k$-partite entangled pure states.

[98] M. Greiter, V. Schnells, and R. Thomale, *The 1D Ising model and topological order in the Kitaev chain,* Ann. Phys. **351**, 1026 (2014).

[99] G. Mussardo, *Statistical Field Theory, An Introduction to Exactly Solvable Models in Statistical Physics* (Oxford University Press, New York, 2010).

[100] B. A. Bernevig and T. L. Hughes, *Topological Insulators and Topological Superconductors*, (Princeton University Press, Princeton, NJ, 2013).

[101] M. V. Berry, *Quantal phase factors accompanying adiabatic changes*, Proc. R. Soc. Lond. A **392** 45 (1984).

[102] I. Mondragon-Shem, T. L. Hughes, J. Song, and E. Prodan, *Topological Criticality in the Chiral-Symmetric Aiii Class at Strong Disorder,* Phys. Rev. Lett. **113**, 046802 (2014).

[103] M. Wimmer, *Efficient numerical computation of the pfaffian for dense and banded skew-symmetric matrices*, ACM Trans. Math. Software 38, 30 (2012); arXiv:1102.3440.

[104] J. K. Asbóth and H. Obuse, *Bulk-boundary correspondence for chiral symmetric quantum walks*, Phys. Rev. B **88**, 121406(R) (2013).

[105] T. Fukui, Y. Hatsugai, and H. Suzuki, *Chern Numbers in Discretized Brillouin Zone: Efficient Method of Computing (Spin) Hall Conductances*, J. Phys. Soc. Jpn. **74**, 1674 (2005).

[106] P. G. Harper, *Single Band Motion of Conduction Electrons in a Uniform Magnetic Field*, Proc. Phys. Soc. London Sect. A **68**, 874 (1955).

[107] D. Hofstadter, *Energy levels and wavefunctions of Bloch electrons in rational and irrational magnetic fields*, Phys. Rev. B 14, 2239 (1976).

[108] Li-J. Lang and S. Chen, *Majorana fermions in density-modulated p-wave superconducting wires*, Phys. Rev. B **86**, 205135 (2012).

[109] I. Mondragon-Shem, T. L. Hughes, J. Song, and E. Prodan, *Topological Criticality in the Chiral-Symmetric AIII Class at Strong Disorder*, Phys. Rev. Lett. **113**, 046802 (2014).

[110] S. Aubry, G. André, *Analyticity breaking and Anderson localization in incommensurate lattices*, Ann. Isr. Phys. Soc. **3**, 133 (1980).

[111] Y. E. Kraus, Y. Lahini, Z. Ringel, M. Verbin, and O. Zilberberg, *Topological States and Adiabatic Pumping in Quasicrystals*, Phys. Rev. Lett. **109**, 106402 (2012).

[112] G. Roati, C. D'Errico, L. Fallani, M. Fattori, C. Fort, M. Zaccanti, G. Modugno, M. Modugno, M. Inguscio *Anderson localization of a non-interacting Bose-Einstein condensate*, Nature **453**, 895 (2008).

[113] M. Schreiber, S. S. Hodgman, P. Bordia, H. P. Lüschen, M. H. Fischer, R. Vosk, E. Altman, U. Schneider, and I. Bloch, *Observation of many-body localization of interacting fermions in a quasi-random optical lattice*, Science **349**, 842 (2015).

[114] Y. Lahini, R. Pugatch, F. Pozzi, M. Sorel, R. Morandotti, N. Davidson, and Y. Silberberg, *Observation of a localization transition in quasiperiodic photonic lattices*, Phys. Rev. Lett. **103**, 013901 (2009).

[115] M. Segev, Y. Silberberg, and D. N. Christodoulides, *Anderson localization of light,* Nat. Phot. **7**, 197 (2013).

[116] P. W. Anderson, *Absence of Diffusion in Certain Random Lattices*, Phys. Rev. **109**, 1492 (1958).

[117] Ad. Lagendijk, B. van Tiggelen, and D. S. Wiersma, Fifty years of Anderson localization, Physics Today **62**, 24 (2009).

[118] E. Abrahams, *50 Years of Anderson Localization* (World Scientific Book, Singapore, 2010).

[119] A. Nava, R. Giuliano, G. Campagnano, and D. Giuliano. *Persistent current and zero-energy Majorana modes in a p-wave disordered superconducting ring*, Phys. Rev. B **95**, 155449 (2017).

[120] L. Lepori, M. Burrello, A. Trombettoni, and S. Paganelli, *Strange correlators for topological quantum systems from bulk-boundary correspondence*, arXiv:2209.04283.

[121] M. Sekania, S. Plugge, M. Greiter, R. Thomale, and P. Schmitteckert, *Braiding errors in interacting Majorana quantum wires*, Phys. Rev. B **96**, 094307 (2017).

[122] L. Pezzè, and A. Smerzi *Quantum theory of phase estimation*, arXiv:1411.5164. Published in *Atom Interferometry*, Proceedings of the International School of Physics "Enrico Fermi", Course 188, Varenna, pag. 691. Edited by G. M. Tino and M.A. Kasevich (IOS Press, Amsterdam, 2014).