# Peer review of "Robust multipartite entanglement in dirty topological wires"

_SciPost Physics_

## Round 1 · Referee Report · Anonymous (Referee 1) · 2022-5-26

Strengths

The paper gives a solid analysis of a long-range free fermion model in the presence of disorder through the analysis of quantum Fisher information (QFI). The study takes into account various types of disorder and different ranges of the hopping. The authors connect the properties of the QFI with the underlying topology of the free fermion model.

Weaknesses

In my opinion the paper points out that the QFI can detect features of the models. It avoids however in trying to go deeper trying to understand what is the reason behind it. QFI, after all, is related correlation functions. It is therefore natural to imagine that this is the reason why the presence of edge modes may contribute. If this is the only reason, I would feel that the observations made on the QFI are a direct consequences of what we know already on the model studied.

Report

The paper is a careful study of a long-range Kitaev chain in the presence of disorder. The model has been extensively studied in the past. The new ingredient is the analysis of the QFI. While the technical part is described very well, the consequences are not discussed. It is not clear what we learn on the model or on QFI from this paper.

Requested changes

The authors should revise considerably the discussion explaining the importance/novelty of their results.

---

## Round 1 · Referee Report · Anonymous (Referee 2) · 2022-6-1

Strengths

1 Transparent presentation
2 Contextualization

Weaknesses

1 No original results
2 Method (multipartite entanglement) might not readily generalize to other more complicated topological phases

Report

To publish or not publish this manuscript in SciPost relies on the assessment to which extent truly new results are a prerequisite for publication. I cannot exclude the possibility that multipartite entanglement, as the authors propose, could become highly relevant in the future - this would speak in favor of publication. At today's stage, however, the amount of original results obtained through multipartite entanglement for the Kitaev chain, as communicated in the manuscript, is very limited. In total, I wish to leave this part of the assessment to the editor-in-charge.

Requested changes

Maybe the authors would like to consider exploring more aspects of entanglement in the Kitaev chain. 1. For instance, the Kitaev chain relates to the Ising model via Jordan Wigner transformation, which is sui generis non-local: (for a later discussion see e.g. https://arxiv.org/abs/1402.5262) How does multipartite entanglement depend on the basis in which the Schmidt decomposition is performed? 2. The authors talk about parametric variations of the chemical potential mu in their article. Could they also just simulate the evolution of entanglement under braiding which would be performed through the variation of mu? (see e.g. https://arxiv.org/abs/1703.03360)

---

## Round 2 · Referee Report · Anonymous · 2022-12-17

Strengths

The new version is improved as compared to the original submission.

Weaknesses

I still feel that some of the critiques present in the reports was not addressed adequately. In particular

- While in the short-range case the connection between the topological properties and the scaling of the QFI information are well described, in the long-range case the corresponding analysis is less clear. This is already present in the abstract where this one-to-one relation is mentioned only for large \alpha. Is it possible to make this kind of conclusion also in the generic case?

- As it stated in the introduction, the determination of topological properties through the QFI may be advantageous in those cases where the computation of topological invariants is difficult. However, as far as I understand, the scaling of the QFI to determine the topological properties of a free fermion system is a guess at present. It is very reasonable that QFI detects critical behaviour. Something I am not able to easily detect from the present paper is the possibility to also distinguish topological properties from the scaling of the QFI.

- I am not sure I understand why the computation of correlation function is difficult in free fermion systems with disorder.

Report

The topic and the nature of the results match the scope of the journal. The paper however, in my opinion, needs still some revision.

Requested changes

The new version should contain an answer to the points raised in the report

---

## Round 2 · Author Response

Dear Editor,

please, find enclosed a new version of the manuscript “Robust multipartite entanglement in dirty topological wires” that we resubmit for publication in SciPost Physics.

We thank both Referees for reading our manuscript. While they have appreciated the analysis reported in our work and found the presentation quite clear, they do not recommend the publication because of the apparent lack of novel strong results.

Let us clarify what is novel in our manuscript.

Existing studies on the inhomogeneous Kitaev chain have mainly focused on short-range pairing where topological phases are identified by topological invariants. However, this approach fails completely for the inhomogeneous long-range pairing model, where the use of common invariants is not justified: for instance, the phase diagrams in Phys. Rev. Research 4, 023144 (2022), -- Ref. [65] of the revised manuscript -- presents regions where the identification/characterization of the topological phases is not possible, because of unavoidable numerical noise associated to the vanishing small energy gap.

In our manuscript, we approach the inhomogeneous Kitaev chain by studying multipartite entanglement, identified by the quantum Fisher information (QFI) - a tool well developed in quantum information, with a primary role in quantum sensing and metrology. This approach was introduced times ago by us for the analysis of the clean Kitaev model, see Phys. Rev. Lett. 119, 250401 (2017), Ref. [33], and has attracted the interest of of the community, including the possibility to extend the study beyond the Kitaev chain case, such as in the Phys. Rev. Res. 3, 013148 (2021), Ref. [28] (where a even genuinely topologically ordered system is considered), but still in homogeneous models.

First, we apply our approach to the short-range pairing case with spatial inhomogeneities, showing a perfect agreement with known results: admittedly, the perfect match between topological invariant and the super-extensive scaling of the QFI is already a non-trivial and beautiful result. Interestingly, we are able to extend the approach also in the inhomogeneous long-range case without major numerical complications. We provide phase novel phase diagrams also extending the calculation for genuine (e. g. à la Anderson) disordered cases, beyond the commensurate quasiperiodic modulations of the chemical potential as in Ref. Phys. Rev. Research 4, 023144 (2022).

Besides recovering known results with a new approach (for the short-range case), and predicting novel phase diagrams (for the long-range case), our manuscript meets, in our opinion, at least two SciPost acceptance criteria: i) it opens a new pathway in an existing research direction, with clear potential for multipronged follow-up work; ii) provides a novel and synergetic link between different research areas: multipartite entanglement and topological matter, in our case. Will the QFI approach be successful to describe other topological systems? Is large (superextensive) multipartite entanglement a unifying aspect of topological matter? These are ambitious questions that go beyond the present work. Nevertheless, our manuscript will certainly trigger further research.

Please, find attached a reply to each comment raised by the Referees, as well as a new version of the manuscript. We are confident that the novel version of the manuscript is suitable for publication in SciPost Physics.

Yours Sincerely,

Luca Lepori and Luca Pezze

---

## Round 2 · List of Changes

Reply to comments by the First Referee.

We thank the Referee for reading the manuscript and for the comments that have stimulated us to improve our work.

> Weaknesses
> 1) No original results

We respectfully disagree with the Referee. In our manuscript, we approach the inhomogeneous Kitaev chain by studying multipartite entanglement identified by the quantum Fisher information (QFI) - a tool well developed in quantum information, with a primary role in quantum sensing and metrology. This approach was introduced by us in Phys. Rev. Lett. 119, 250401 (2017), Ref. [33], for the analysis of the clean Kitaev model. This work has attracted the interest of the community, including the possibility to extend the study beyond the Kitaev chain case, such as even models with genuine topological order, as in the recent Phys. Rev. Research 4, 023144 (2022), Ref. [65], but still in homogeneous models.

First, we apply our approach to the dirty short-range pairing case, showing a perfect agreement with known results: admittedly, the perfect match between topological invariant and the super-extensive scaling of the QFI is already a non-trivial and beautiful result. Therefore, the QFI appears a valid phase (transition) indicator, also in the presence of different types of space inhomogeneities.
We then consider the long-range case, where there is not a definitive approach to identify and characterize topological phases when space inhomogeneities are present, due to difficulties in defining a topological invariant. In contrast to existing studies, our approach provides reliable diagrams not only for commensurate quasiperiodic modulations of the chemical potential (as in Phys. Rev. Res. 3, 013148 (2021), Ref. [28]), but also for genuine (e. g. à Anderson) disordered cases. These results are new in literature.
On the conceptual side, our study testifies, for the first time to the best of our knowledge, the robustness of multipartite entanglement (as witnessed by the QFI) against inhomogeneities also with long-range pairing.

To emphasize these aspects of our work, we have added the sentence “The present work contributes to establish ME as a central quantity to study intriguing aspects of topological systems and testifies its robustness against spatial inhomogeneities.” at the end of the abstract.

In the introduction we have added the sentences: “In the SR case, we observe a perfect agreement between the super-extensive scaling of the QFI and non-trivial topology phases identified by topological invariants [14-16].”; “In the LR case, where topological invariants are generally difficult to be defined, especially in the absence of translational invariance, our methods are particularly relevant since they provide a clear characterzation of the phase diagram and the unprecedented identification of LR phases.” ; and “On the general conceptual side, our study testifies, for the first time to the best of our knowledge, the robustness of spatial multipartite entanglement against spatial inhomogeneities.”

> 2) Method (multipartite entanglement) might not readily generalize to other more complicated topological phases

The (clean) Kitaev chain is one of the most important models of symmetry-protected topological order and its extension to long-range pairing has attracted large interest, being a remarkable example of solvable long-range model. Our manuscript reports a unifying approach valid for short- and well as long-range pairing, including different interesting inhomogeneities. No other approaches to this celebrated model has proved a comparing level of success.

We agree with the Referee that the possible generalization of our approach to other more complicated models is not straightforward, and remains an open question in the literature. However, i) conceptually, we do not see any major obstacle for such an extension, and ii) the reported analysis will certainly trigger the community working at the interface of quantum information and many-body theory toward such an extension.

> Report
> To publish or not publish this manuscript in SciPost relies on the assessment to which extent truly new results are a prerequisite for
> publication. I cannot exclude the possibility that multipartite entanglement, as the authors propose, could become highly relevant in > the future - this would speak in favor of publication. At today's stage, however, the amount of original results obtained through
> multipartite entanglement for the Kitaev chain, as communicated in the manuscript, is very limited. In total, I wish to leave this part
> of the assessment to the editor-in-charge.

We have already commented above about the original results reported in our manuscript.

> Requested changes
> Maybe the authors would like to consider exploring more aspects of entanglement in the Kitaev chain.
>
> 1. For instance, the Kitaev chain relates to the Ising model via Jordan Wigner transformation, which is sui generis non-local: (for a later > discussion se e.g. https://arxiv.org/abs/1402.5262). How does multipartite entanglement depend on the basis in which the Schmidt > decomposition is performed?

We thank the Referee for suggesting the interesting work of Greiter et al., which is now cited as Ref. [99] when introducing the Jordan-Wigner transformation.

Multipartite entanglement (as well as the common bipartite entanglement) certainly depends on the identified parties, namely the decomposition of the system's Hilbert space, but not on the basis of such subspaces. Furthermore, the QFI, being a susceptibility to a given dynamical evolution, depends on an additional operator J generating, in principle, such an evolution, see Eq. (3). It is clear that are “good” and “bad” operators J: the good ones are those for which the QFI is large. In our case, we exploit the Jordan Wigner transformation, as an educated guess about the choice of operator J, and optimize over a wide class of pseudo-spin operators. The identification of J for other topological models is the main obstacle for the calculation of the QFI in general models and remains an open question in the literature. However, we believe that emerging numerical approaches, such as reinforcement learning for instance, can help to circumvent this problem: this is still a subject of research by us and the identification of J beyond an educated guess remains beyond the scope of this manuscript.

We have added the following sentence after Eq. (11): “Multipartite entanglement witnessed by the QFI Eq. (4) does not depend on the basis of the multipartite decomposition but it depends on the choice of operator J considered. In this respect, Eq. (5) is chosen as an educated guess, directly suggested by Jordan-Wigner mapping [34, 63].”

> 2. The authors talk about parametric variations of the chemical potential mu in their article. Could they also just simulate the
> evolution of entanglement under braiding which would be performed through the variation of mu? (se .g.
> https://arxiv.org/abs/1703.03360)

The question by the Referee is extremely interesting but it goes well beyond the scope of this manuscript. We have cited the work by Sekania et al., Ref. [120] in the conclusions, leaving the investigation of evolution of entanglement under braiding as possible future perspective. We have added the sentence “The study of the QFI presented here can be also extended to analyze the dynamical evolution of entanglement, e.g. under braiding [122].”

>>>>>>>>>>>>>>>>>>>>>>>>>>>>>>>>>>>>>>>>>>>>>>>>>>>>>>>>>>>>>

Second referee

> Weaknesses
> In my opinion the paper points out that the QFI can detect features of the models.
> It avoids however in trying to go deeper trying to understand what is the reason behind it. QFI, after all, is related correlation
> functions. It is therefore natural to imagine that this is the reason why the presence of edge modes may contribute.
> If this is the only reason, I would feel that the observations made on the QFI are
> a direct consequences of what we know already on the model studied.

We respectfully disagree with the Referee. In the commensurate case, as well as in the case of Anderson disorder, the calculation of the QFI is performed with close boundary conditions, therefore clearly excluding a relevant role of edge modes. In this way, the main qualitative feature for the QFI result from the bulk structure, in a similar manner than topology: invariants for topological insulators and superconductors are primarily calculated in the bulk and assuming translational invariance.

We emphasize that the analysis of correlation functions is very difficult and inconclusive in disordered systems: in this case, we have observed that correlation functions show a rather complicated staggered structure that prevents extracting clear trends, especially in systems of relatively small size. For this reason, correlation functions have not been used - to the best of our knowledge - to analyze the systems we consider in this work. For short-range pairing, authors have preferred the calculation of topological invariants and have tried to extend them for long-range pairing, without complete success.

On the contrary, our result show that a key information about the system is contained in the scaling of the QFI with the systems size, even in the presence of inhomogeneities. For pure states, the QFI reduces to the variance and is, for pure as well as mixed states, a witness of multipartite entanglement. We extract, with a reasonable numerical effort, clear power-law scalings of the QFI that identify/characterize topological phases.

We have added the following sentence in the conclusions: “Similarly, the analysis of correlation functions is notoriously difficult for inhomogeneous systems, as they can show complicated staggered structure, while the QFI is characterized by clear power-law scalings.”

> Report
> The paper is a careful study of a long-range Kitaev chain in the presence of disorder. The model has been extensively studied in the
> past. The new ingredient is the analysis of the QFI. While the technical part is described very well, the consequences are not discussed. > It is not clear what we learn on the model or on QFI from this paper.

We agree that the short-range Kitaev model has been extensively studied in the past. In this case, we believe that showing a perfect agreement between topological invariant and the super-extensive scaling of the QFI is already a novel, non-trivial and beautiful result. Yet, we respectfully disagree about the long-range inhomogeneous Kitaev model being well known. On the contrary, attempts to identify topological phases in this case have been recently reported in the literature but were not completely successful. For instance, the phase diagrams in Phys. Rev. Research 4, 023144 (2022), Ref. [65], present black regions where the identification and characterization of the topological phases is not possible because of numerical noise associated to the small energy gap. As a matter of fact, currently, there is no other approach to detect and characterize topological phase in the Kitaev chain with long range pairing and in the presence of inhomogeneities. Our approach overcomes these difficulties and overall, our study indicates that the QFI is a valid indicator of topological phase transitions, also in the presence of different types of disorder and long range pairing. Even more relevant, our study testifies, for the first time at the best of our knowledge, the robustness of multipartite entanglement against disorder.

> Requested changes
> The authors should revise considerably the discussion explaining the importance/novelty of their results.

We have revised the introduction and the conclusions: we believe that the novelty and importance of our work is now well emphasized in the manuscript.

---

## Editorial Decision

awaiting_resubmission